# Posture and Pull Pressure by Horses When Eating Hay or Haylage from a Hay Net Hung at Various Positions

**DOI:** 10.3390/ani12212999

**Published:** 2022-10-31

**Authors:** Samantha Hodgson, Pam Bennett-Skinner, Bryony Lancaster, Sarah Upton, Patricia Harris, Andrea D. Ellis

**Affiliations:** 1Royal (Dick) School of Veterinary Studies, University of Edinburgh, Easter Bush, Midlothian, Edinburgh EH8 9YL, UK; 2School of Animal, Rural and Environmental Science, Brackenhurst Campus, Nottingham Trent University, Southwell NG25 4QF, UK; 3Waltham Petcare Science Institute, Freeby Lane, Waltham-on-the-Wolds, Melton Mowbray, Leicestershire LE14 4RT, UK; 4Unequi Ltd., 12 Bridgford Street, West Bridgford, Nottinghamshire NG2 6AB, UK

**Keywords:** equine, haynet, posture, intake behaviour, pull pressure

## Abstract

**Simple Summary:**

Haynet use is a widespread management approach, but they can impose an unnatural foraging strategy on horses. Previous studies have highlighted that when horses eat from haynets, a strong force is exerted that can lift the haynet up and away from the wall. In this series of studies we assessed these forces and neck angles under varying conditions using video observation and pressure gauges.

**Abstract:**

These studies assessed the pressure forces exerted by horses to extract forage from haynets. Study 1 measured horse posture and pressure in Newtons (10 N = 1 kg Force) exerted on haynets when feeding from either a single (SH) or double layered (DH) haynet (3 kg Hay), hung low or high. Mean and maximum pull forces were higher for the DH vs. SH (DH: 81 ± 2 N, max 156 N; SH: 74 ± 2.9 N, max 121 N; *p* < 0.01). Horses pulled harder on low (max pull 144 ± 8 N) compared to high (109 ± 8 N; *p* < 0.05) hung haynets. Mean maximum angles (nose-poll-withers) recorded were 90° ± 9 for SH and 127° ± 10 for DH (*p* < 0.01). Study 2 was a latin square design measuring forces exerted by 10 horses when eating from haynets (6 kg fill) with hay or haylage and attached to the wall at single or double points. Pull pressures were significantly higher when eating haylage compared to hay (mean: 7.5 kg vs. 2 kg and max: 32 kg versus 12 kg, respectively, (*p* < 0.001). Forage type and fracture properties had the greatest effect on apprehension rates of hay from haynets. In this study, the majority of force exerted when eating from haynets was below 70 N for hay and for haylage 50% of pulls were higher than 50 N with 8% of pulls above 200 N.

## 1. Introduction

Feed intake behaviour is an ethological requirement and feral or pastured horses are motivated to forage for a minimum period of 12 ± 2.5 h/24 h [1,2]. Many horses are not fed ad libitum forage, either to prevent obesity and related conditions or due to a continued assumption that optimum performance cannot be achieved when feeding high levels of forage [1]. Studies where domesticated horses are offered feedstuffs that limit feeding times, due to reduced volume or fast ingestion rate, show that horses will either ingest excessive amounts [3,4] or find alternative ways to carry out foraging related behaviours [2,5,6]. Therefore, it was concluded by Harris et al., that healthy horses should ideally be provided with the opportunity to perform foraging behaviour for a minimum of 8 h per day [1,2].

Haynets are commonly used to present forage to horses with owners reporting their effectiveness at reducing the amount of wasted forage and limiting parasitic contamination [7]. With the increased understanding of the ethological requirement to chew for horses, haynets have been researched extensively as a tool to increase forage intake times and thus keep horses busier for longer [3,4,6,8] (Figure 1).

These studies have highlighted that haynet type (hole sizes and/or double layered haynets), fill size and forage type may affect feed intake times and Ellis et al. [3,6] pointed out that there was a strong effect of haynet hanging height and haynet type on posture and visible force necessary to extract the hay. In addition, different types of forages have different fracture properties, changing the pressure required to extract them.

The aims of the studies reported here were to assess the effect of haynet type, forage type and haynet hanging options on the pressure exerted to pull out different types of forage and to have a preliminary look at posture when eating from the haynets.

## 2. Materials and Methods

A cross over study using 6 horses and a latin square design study using 10 horses were carried out. The first study aimed to assess head-neck posture and the force exerted on haynets when hung at different heights as either single haynets or doubled up haynets (Study 1—haynet type). The second study (Study 2—forage type) assessed pressure when presenting haynets with two different hanging methods (single attachment and double attachment) and with two types of forage (haylage vs. hay). Both studies received ethical approval by the R(D)SVS Veterinary Ethical Review Committee and Study 2 was approved by the R(D)SVS Veterinary Ethical Review Committee and ARES Ethics Committee, Nottingham Trent University. All horses were acclimatized to the treatments and no welfare issues arose.

### 2.1. Study 1—Haynet Type

#### 2.1.1. Study Design and Horses

Six horses (3 mares & 3 geldings; bodyweight 460.6 kg ± 65.29 s.d.; age 9.8 years ± 3.8 s.d.) were used in a cross-over design (Single layered haynet—SH × Double layered haynet—DH) consisting of 6 days adaptation followed by 2 phases of 4 days of data collection (test sessions) each. Within each phase two haynet hanging heights were tested. Horses were initially randomly assigned to phase groups (SH or DH per cross-over phase) and haynet starting height alternated during each test session. Horses were worked lightly as leisure horses but during data collection period horses were not exercised.

During the adaptation period horses were acclimatised to the measurement box and to feeding from that phase’s haynet type for 6 days for 30–40 min/day. A test period lasted for 30 min which was split into two haynet hanging heights (15 min each): attachment height—top of haynet—for low hung haynets (low) was + 2.54 cm above withers and for high hung (high) haynets was withers height + 30 cm (Table 1).

Each measurement combination (SHlow; SHhigh; DHlow; DHhigh) was therefore measured 4 times for 15 min per horse.

#### 2.1.2. Feeding Management and Hay Treatment

Prior to the start, dental condition was assessed and if necessary adjusted and horses were checked for absence of back and neck pain by a veterinarian. All horses were out on pasture which was scarce and of low quality for at least 12 h/day. Horses were fed additional early cut meadow hay outside or when in box stalls and received one supplementary feed per day (varying times dependent on day, 07:00 h on testing days). The same batch of hay was used throughout the study and it was relatively brittle and short cut due to a very dry and hot season during growth.

Forage was withheld for 45–60 min prior to commencement of measurements. For each measurement period 3 kg of hay was presented in small hole hay nets with 2.5 × 2.5 cm holes (Shires Haylage Net) single layered (SH) or double layered (DH) as described in Ellis et al. [6].

#### 2.1.3. Measurements

Video observation and pressure measurements took place and amount eaten during the observation was recorded (gram). Together with the haynet the starting weight on the pressure gauge was 40 N (~4 kg). Horses were positioned along the backwall of the stable in front of the haynet.

Haynets were attached to the gauge unit of a force meter (M&A Instruments HF: 50–5000 N Digital Force Gauge Meter Tester Push Pull Gauge with External Sensor). The gauge was attached to the wall, a data cable transferred information to the pressure sensor unit from where a cable was attached to a laptop behind the stable wall. Forces applied to the net were recorded in Newtons (N) (1 N = 0.10197 kg pull force, e.g., 50 N = 5.1 kg). The gauge was calibrated at the factory and re-tested in a short pilot study and recorded from a lower limit of 50 N. Push pressures were also recorded as negative pressure (−N). These were discarded before assessing mean pressure exerted.

Horses were video recorded, with a camera (GoPro Hero3+) mounted parallel to the horse 2 m from the starting shoulder position of the horse at point of shoulder height. Height of camera was therefore kept identical for all horses. Horses were kept in the position parallel to the wall and put back in place when moving to minimize changes in camera to horse difference which may affect angle recording slightly.

For measurement of angles, white adhesive dots were placed on the horse at the point of the cheekbone closest to the nostril (nose), at the base of the ear, on the neck half way to point of withers, the point of withers and the point of shoulder. Two angles were recorded: poll angle (nose-poll-withers) and wither angle (neck-wither-point of shoulder) (Figure 2; Sports-movement angle evaluation by ©2019 CoachMyVideo). A still photo was used to ensure exact attachment points of white dots for each horse between tests.

From this, exact time spent eating (minutes), posture (horse’s engagement with the net/Figure 2) and max head-neck angles were assessed using still pictures taken when head was at highest point after a bite-pull/fling and mean angles were taken from still pictures taken every 10 s from the recording for 5 min.

An ethogram for eating and posture was devised from the observations identifying time and counts for: eating; biting; bite-fling (contact maintained by horse with net being lifted from the wall) and flinging (net lifted upwards with removal of contact by horse) (Figure 2) recorded from continuous video observations.

#### 2.1.4. Data Analysis

The pressure gauge recorded all pressures above 50 N at a rate of 10–11 measures/per second, leading to an average of 17,950 pressure points per observation. Data was reviewed and identical recordings within 0.5 split seconds were omitted from the analysis together with extreme outliers and push data (negative measures). Apart from maximum, mean and median pull-forces, the number of pulls above and below 50 N were calculated for comparison.

Data was analyzed with SPSS 19 (2016). Quantitative data was analyzed for normal distribution using the Shapiro–Wilks Test. A multivariate Anova test was carried out to test for interaction between height and haynet type. Results showed no interaction between these for head angles or pull pressures (Manova). Therefore, effect of treatments was analysed separately with pairwise comparison (*t*-test/Wilcoxon sign test). For correlations Spearman Rank Correlations were applied. The significance level was set as *p* < 0.05. Data is reported as means ± standard error unless indicated otherwise.

### 2.2. Study 2

Study 1 had some limitations, in particular: the gauge did not record detailed pressure below 50 N and therefore using only 4 kg (40 N) of haynet weight limited accurate lower readings—omitting pressure readings below 20 N (assuming the haynet continued exerting a weight of at least 30 N as less than 1 kg hay was consumed during the observation period); the video images of ‘flinging’ behaviour had to be matched to pressure readings and ‘false’ high readings of the ‘haynet’ falling into the gauge from a great height were removed ‘manually’. The bite-fling behaviour observed in the Study 1 confirmed frustration behaviour which led us to explore additional haynet attachments in Study 2, which may help the horse.

Therefore, Study 2 measured the effect of two types of forages (hay [HY] or haylage [HG]) presented in the same haynets but at 6 kg fill, as well as additional hanging attachments (single top attachment as in Study 1 [s], and double attachment of haynet—top and bottom [d]), on forces exerted by 10 horses when eating from haynets.

#### 2.2.1. Study Design and Horses

Ten horses were used (9 geldings, 1 mare; bodyweight 520 kg ± 35 s.d.; age 7–16 years) in a repeated measures Latin Square design (4 treatment combinations × 4 repeats = 160 test periods), in which horses were randomly assigned to two groups and each group participated twice per day in ‘blocks’ organised across 8 days of data collection (Table 2). Each horse participated in 16 × 10–15 min data collection periods and these were equally split between morning and afternoon sessions for each horse.

Horses were located at Nottingham Trent University, Equestrian Centre, Southwell, Nottinghamshire, and were selected due to their experience of getting both hay and haylage at some point in the previous few months for a minimum of 2 weeks with the same haynets as in Study 1. Therefore, horses were already acclimatised to both forages and haynets used (2.5 × 2.5 cm holes; Shires Haylage Net) and all had undergone routine dental treatment prior to the study. Horses received their normal current daily diet of either hay or haylage together with two supplementary feeds at 07:00 h and at 17:00 h. Horses were used in the riding school. Both hay and haylage were from mixed seeded pastures containing ryegrass, foxtail and some timothy grasses.

#### 2.2.2. Measurements

Horses were stabled throughout the day and led to the data collection box to which they had been acclimatised previously, as it is normally used as a washing and grooming box. The aim was that horses were not ridden for 1 h prior to a measurement session, however this was not always achieved and therefore effect of exercise prior to tests was assessed.

For each measurement period a 6 kg haynet of hay or haylage was presented (2.5 × 2.5 cm holes; Shires Haylage Net). Haynets were hung from a height of 30 cm above the height of the withers, in line with Study 1’s ‘higher’ hanging treatment, as results showed that less pressure was required to eat from this position. Haynets were weighed out after each session to calculate the quantity eaten, intake rate (g/minute), intake times (minutes/kg) and apprehension rate (g/bite). An indirect measurement of bites (counts/kg) relating to ‘pull data’ from the haynet was recorded (see data analysis).

To measure pressure exerted between horses’ teeth and the haynet, a pressure gauge was inserted between the haynet attachments and the wall attachments as in Study 1. A digital force gauge unit from Kern & Sohn GmBH was used (Sauter FL-M, FL-2K, 0–2500 N—no minimum N pressure limit). The gauge attached data via a cable to a secure laptop and gauge calibration was checked between each session (as Study 1). For double attachment a rigging system was attached to the gauge unit at the top via the bottom haynet attachment to amalgamate pressure readings from both attachments.

#### 2.2.3. Data Analysis

Data from the gauge recorded pressure every 0.12 s (8/sec) directly into the computer software program provided by the manufacturer. Data was uploaded into MATLAB (TheMathWorks, 2012), which was programmed to correct data to a baseline of 0 (accounting for initial 6 kg gross weight of forage + haynet) and to extract data points while omitting negative readings (push pressure) and the readings from the haynet ‘dropping’ from height once forage was extracted. Outliers due to haynets dropping into the ‘gauge’ from a height on the single attachment could be identified easily as the pull pressure increased slowly in a curve (consistent pressures for more than 10 data points) with a sudden short drop in pressure (haynet falling—0 to less than 5 N) followed by a very high and short pressure reading (haynet ‘lands’—less than 10 data points—the ‘landing’, Figure 3). Therefore, parameters were programmed that extracted data of a minimum of 5 N (negating negative readings) and a minimum of 10 data points (1.12 s) apart. The rest of the positive data points were identified as ‘pulls’ and also used to record ‘bites’ in terms of frequency of occurrence.

Data was statistically analysed using SPSS (IBM Corp. Released 2016. IBM SPSS Statistics for Windows, Version 24.0. Armonk, NY, USA: IBM Corp.) and the significance level was set at *p* < 0.05. A Kolmogorov–Smirnov test was used to test for normal distribution. Due to a strong interaction effect of attachment and forage type (Manova), the effect of forage on pressure readings was measured separately within each attachment treatment (Anova). Unless otherwise defined, results are reported in mean ± standard error (SEM) and forage quantities are given in wet matter (WM = as fed), and pressure readings are presented in Newtons of kg (10 N = 1.0197 kg).

## 3. Results

### 3.1. Study 1

After video evaluation and manual removal of extreme outliers due to ‘haynet drop’, the maximum pressure recorded was 378 N (range 53–378N) which is equivalent to lifting 38 kg. The mean maximum pull forces were significantly lower for single haynets compared to double haynets (SH: 114 ± 8.2 N, DH: 157 ± 7.7 N; *p* < 0.001, n = 12, Wilcoxon Figure 4). The mean pull forces >50 N were also significantly higher for the double haynets versus single haynets (SH: 64 ± 2.5 N; DH: 80 ± 2.3 N, *p* < 0.01; Wilcoxon).

There was a significant effect of hanging height on pull pressures above 50 N. Horses pulled significantly harder on low hung compared to high hung haynets (max pull LH: 152 ± 10; HH:106 ± 11; *p* < 0.05; n = 12; paired *t*-test).

A greater amount (trend) of ‘haynet flinging’ occurred on the double layered haynets (2.7 ± 0.8 times/15 min) compared to the single haynets, irrelevant of height (0.7 ± 0.2; *p* = 0.06; Mann–Whitney). The mean maximum poll angles were smaller for single haynets (90° ± 9) than for double haynets (127° ± 10; *p* < 0.01; n=12; paired *t*-test), irrelevant of hanging height, highlighting a greater change in posture on double haynets (37° greater angle) due to the higher resistance of the net to ‘release’ its forage (Figure 5).

There was a great variation between horses for high hanging double haynets. No effect of any treatments on wither angles (poll-wither-shoulder) was seen (Figure 5).

### 3.2. Study 2

Of the 160 test periods, 155 test periods were included (HYs = 39, HYd = 40, HGs = 38 and HGd = 38) in the analysis. Five test periods were excluded due to equipment failure and refusal to eat due to external distractions on the yard. The mean testing time of test periods was 10.28 ± 0.11 min.

There were significantly higher mean, maximum and median pressures when pulling from the double attachment haynets compared to single attachment for both forages (Mean *p* < 0.05; Max *p* < 0.001, hay F = 52.8, haylage F = 108, Anova, Figure 6).

For single attachment data, there were significantly higher mean (Mean HG 74.2 ± 16.5 N, HY 20.5 ± 5.6 N, F = 95.3, *p* < 0.001, Anova), maximum and median pull pressures when eating haylage compared to hay. Comparing pulls according to pressure ranges also showed significantly higher occurrence of pulls >50 N for haylage and significantly higher numbers of pulls below 30 N (*p* < 0.001) for hay (Figure 7).

For the double attachment data, there also were significantly higher mean, maximum and median pull pressures when eating haylage (Mean HG 192.8 ± 29.3 N, HY 39 ± 10.8 N; *p* < 0.001, mean F = 242.5, max F = 190.1, med F = 102.1, n = 10, Anova), with significantly higher numbers of pulls above 100 N (*p* < 0.001, 100–200 F = 36, >200 F = 253, Anova) for haylage and significantly higher numbers of pulls below 100 N (*p* < 0.001, 30–50 F = 87, 20–30 F = 165, <10 F = 5.2, Anova) for hay.

The overall distribution of all pull pressure data illustrates that higher pressures were exerted when eating haylage compared to hay (any attachment, Figure 6 and Figure 7), but highest pressures were exerted when on a double attachment (any forage, Figure 6).

#### 3.2.1. Intake Behaviour

There was a significant difference in all individual feed intake behaviour measures (bites/minute, grams/bite and intake time in minutes/kg) between forage types (*p* < 0.001, b/m F = 82, g/b F = 65, m/kg F = 61, Anova, Figure 8) irrelevant of attachment type (no interaction). Attachment type had no effect on intake behaviour parameters.

The higher pressure exerted to extract haylage compared to hay (Figure 6), resulted in a greater amount extracted per bite (g/bite, Figure 9 dark boxplots) while performing less bites/minute (Figure 9 light boxplots), leading to the faster (lower) intake time (Figure 8). Attachment type did not affect this significantly.

#### 3.2.2. Other Variables

It was not possible to control for the forage conditions or the method of presentation in the stable that the horse’s experienced prior to the test period as this was the forage and normal feeding method determined by the yard manager for each horse. Therefore, prior conditions were noted for analysis. Some differences could be highlighted but no direct interactions/effects from forage type prior to test period were found on feed intake behaviour or pull pressure (Manova).

The intake time/kg wet matter for hay was significantly higher (so slower intake rate in g/min) when hay was in the stable prior to testing (53 ± 4 min/kg, n = 27), than when there was no forage (43 ± 1.4 min/kg, n = 44, *p* = 0.016) or haylage (36 ± 1.2 min/kg, *p* = 0.012, n = 8, Bonferroni) present prior to testing (Figure 10—lower case superscripts). The intake time for haylage was slightly higher (strong trend) when haylage was in the stable prior to testing (29 ± 1.7 min/kg, n = 11), than when there was hay in the stable prior to testing (26 ± 0.7 min/kg, *p* = 0.05, n = 28, Bonferroni, Figure 10—upper case superscripts).

When the forage treatment type was hay, mean pull pressure was significantly higher when haylage (46 ± 6.8 N, n = 8) had been in the stable prior to testing than when hay (27 ± 2.1 N, *p* = 0.004, n = 27) or no forage (29 ± 2.1 N, n = 44, *p* = 0.006, Bonferroni) had been in the stable prior to testing (*p* < 0.01, F = 5.9, Anova, Figure 11)

Exercise of horses in proximity to testing periods showed no significant effect on feed intake behaviour or pull pressure (exercised: 24%; not exercised: 76%, Manova). Equally, the time of day had no significant effect on feed intake behaviour or pull pressure (Manova).

## 4. Discussion

The aim of these two independent studies was to take a first look at the pressures exerted by horses when eating from a haynet and how these are influenced by forage type, forage fill, hanging height and horse factors. A direct comparison between studies has limitations as the second study was built on the experience gained from the first study. As such the first study focused on gauging maximum pressures exerted while the second study tries to look much closer at the overall mean pressures by use of a pressure gauge which also includes readings down to 0 N.

### 4.1. Effect of Hanging Height on Posture and Bite Force (Study 1 in Context of Study 2 Results)

As this was the first study of its kind, the reliability of pressure readings was assessed for Study 1. The mean % cv irrelevant of treatment and height for pressure gauge data ranged from 19 to 26% cv between measures and from 13 to 30% cv within repeated measurements per horse. Average % cv for all pressure measures was 22 which is acceptable for reliability of overall results although a lower level would be preferable. However, there will naturally be a greater variation for pressure recordings, according to the chosen position on the haynet when taking a bite (i.e., top of haynet versus lower down).

One of the limitations of Study 1 was that bites resulting in pressures below 5 kg (50 N) were not recorded. Average pull pressure for pulls recorded above 50 N equated to 7–8 kg and this does not seem of high concern in relation to the strength of horses’ teeth and neck muscles, especially as pulls below 50 N (5 kg) were not recorded with this gauge, inflating the mean pressure values. Data from Study 2, where there was no lower detection limit for the gauge, indicated that over 70% of bites recorded for hay (Figure 7) were below 20 N and therefore the mean pressures (>50 N) in this Study (DH: 81 ± 2 N, SH: 74 ± 2.9 N, *p* < 0.01) may be higher than the actual overall pressure. This is confirmed by the lower mean in Study 2 for hay (20 ± 1.2 N) with the caveat that two different hays were used in each study, although both were quite brittle due to hot summer harvests.

However, for relevance to the horse’s dental health the maximum pressures and the occurrence of higher pressures were of special interest (DH 156N; SH: 121 N; *p* < 0.01) and these correspond with results from Study 2 for hay (SH 120 ± 14 N). Hongo and Akito [9] used different transducers on an artificial sward to estimate pressure used by grazing horses when biting the sward and although the methodology was very different, they measured a mean maximum biteforce of 141 ± 11 N for horses which is comparable to the maximum pressures measured on the double haynet in Study 1 but lower than that for haylage in Study 2.

That increased pressure with double haynets was expected, but the level of increase overall was not extreme at around 25%. This was much lower than the mean maximum pressure exerted when eating haylage from a single haynet in Study 2 (307 ± 18 N). The hay used in Study 1 was short cut and fractured easily as a result of typical growing conditions in the region (hot, irrigation dependent), resulting in lots of small repeated ‘grazing’ like bites at times. Such hay would benefit from using double haynets, leading to a slower intake rate while not considerably increasing pressure exerted. Though based on these results, using double haynets for forages which are difficult to extract, such as the haylage from study 2, should be avoided for prolonged periods.

The very small ‘gain’ in g/bite (apprehension rate) may have led to some of the frustration behaviour described as ‘flinging’ of the net, the occurrence of this increased from a mean of 1.9 times per 15 min observation for single to 2.7 times for double haynets but variation between horses was very high (SEM 39). Possibly temperament and previous experience may have played a role. LeSimple et al. [10], found stabled riding school horses fed from elevated feeders to have higher levels of back disorders but too many variables could have affected results. Ellis et al. [6], used a more mature hay and reported a strong effect on food intake behaviour when eating from double haynets, involving ‘flinging’ of nets. This was also observed in Study 1 and the poll angles were high at times, with some recorded above 189° with the muzzle far above the ears of the horse (Figure 2).

Mean maximum poll angles were slightly lower (not significant) on low hanging haynets as horses tended to eat more from ‘above’ (Figure 12). Overall some of the extreme angles recorded have potential to cause musculoskeletal damage to the back and neck [11]. Interestingly, studies measuring the pressure from reins on horses’ mouths showed pressures of up to 60 N [12,13]. Haynet fill will also affect posture and angles considerably and therefore observations over a longer period of time or studies using different haynet fill levels will be useful.

Whereas angle at poll was slightly higher on higher hanging haynets (30 cm above wither), higher haynets reduced the pressure exerted per bite possibly through the gravity of pulling hay downwards out of the net. For this reason the position of 30 cm above withers was chosen for Study 2.

### 4.2. Limitations (Study 2)

There were some limitations in relation to analysis of gauge measurements, which need to be considered when discussing the results of Study 2. Analysis of video recordings has not yet been completed and therefore effect on intake behaviour (bite frequency, gram/bite) is limited to pressure readings. Visual counts of the number of bites per session (number of data points) were made for 20 individual sessions (one for 10 horses per forage, single attachment) and compared with the pressure readings after MATLAB data extraction, which indicated a strong correlation and supported the parameters chosen (*p* < 0.001, Pearson Correlation). However, when matching the correlated data for number of bites (assuming each ‘pull pressure’ is a bite), between the haylage and hay variables, there was an overestimation of the number of bites in the haylage variable and an underestimation of the number of bites in the hay variable. Individual pressure readings are therefore not perfectly representative of real life bites and further visual analysis of video recordings will take place in the next phase of the project, which will also focus on postural assessment as a variable for analysis. Nevertheless, pressure data after MATLAB extraction allowed for the elimination of peaks that occurred due to haynet drop and this fulfilled the aim of measuring pressure exerted by horses teeth during eating from haynets. In Study 1 this had to occur by visually matching up all recordings with pressure graphs.

### 4.3. Effect of Type of Forage and Attachment (Study 2)

Mean pull pressure was 2.6 times higher and maximum pull pressure was 1.6 times higher when eating haylage than when eating hay, and hay led to a 83% increase in bite frequency. This difference was clearly visible during observation and in pressure graphs (Figure 13).

In terms of the congruency of results in pressure readings between studies, these indicate repeatability of the methodology after adjustment of Study 2 results to pressures >50 N (as used in Study 1) (Table 3). As with any study comparison, we are not comparing an exact replication of methods but despite a difference in forages used and haynet fill between our two studies (which may well have influenced results somewhat as highlighted by Ellis et al. [3]), the pressures exerted when pulling hay or haylage from the haynet shows some overall consistency.

The aim of the double attachment was to see if less pressure would be exerted because of haynets resisting being lifted completely into the air as seen in the pilot Study. A higher amount of pressure was recorded when comparing mean pull pressure between studies, though because of the lack of separate low pressure data collected, and the challenges of interpreting double attachment data from Study 2, it’s not possible to draw a conclusion about this specific variable at this point. However, the double attachment results are of some interest in relation to intake rates which will be discussed later.

Mean pull pressures corresponded to 2 kg and 7.5 kg of force, and maximum pull pressures corresponded to 12 kg and 31 kg of force for hay and haylage respectively. The aforementioned study by Hongo and Akimoto [9] found equivalent mean bite pressures for horses of 10–12 kg with 17–30% of bites exceeding 31 kg when eating from constructed swards. Therefore, findings may be within the normal range for foraging behaviour. However, it is not possible to say whether postural differences could have an impact, or whether the methodology used by Hongo and Akimoto (16) was truly reflective of normal grazing. When grazing, horses use their incisors to ‘shear’ as well as pull, which is less likely to occur when eating from haynets.

When eating haylage, 8.7% of bites exerted >20 kg of pressure, with the highest recorded maximum pull pressure being 48 kg. Peaks of this magnitude with regularity could be detrimental to dental health and neck muscles in the long term, with animal models suggesting that repetitive flexion-extensions link more reliably with injury than compression values, but with increasing injury risk when compression is increased within highly repetitive use [14]. The haylage used in Study 2 was visibly longer in chop length and of a later cut with more stemmy material compared to the hay. The additional moisture content may have increased elasticity and most likely influenced fracture properties. Similar to the hay of Study 1, the hay in Study 2 has been grown in a very hot summer with reduced growth rates due to drought. Tensile strength is related to both moisture content, stage of development, and degree of lignification [15,16] so both moisture content, maturity of plant and chop length will have affected differences in pressure required to extract forage from haynets.

When eating hay, it was found that there was a strong trend for more bites per minute for the double attachment compared to single attachment (*p* = 0.051), which was likely the result of a significantly reduced amount of grams extracted per bite (single: 1.9 ± 0.2; double: 1.4 ± 0.1). The double attachment was less maneuverable than the single attachment and this prevented horses from easily accessing larger bites of hay from the full haynet—this may level out over longer term observations. Therefore, horses may have adapted an increased bite frequency in order to maintain a similar intake rate when the extracted mouthfuls were smaller. However, double attachments could have the potential to slow down intake rates, which is particularly desirable for horses fed restricted forage for health reasons [17].

Forage type had a strong effect on intake rates, with significantly more bites of significantly smaller apprehension rates when eating hay, leading to a 63% slower intake rate. On average an additional 17 min would be required to consume 1 kg of hay (44 min/kg) compared to 1 kg of haylage (27 min/kg) from these results. Only short term intake rates were recorded in these studies, compared to 1 h observations in Ellis et al. [3,6], where the same haynets were used and recorded intake times of 25–28 min/kg Wet Matter (WM) hay (mature cut, long hay) were found, which are similar to findings in the present Study for haylage, but not hay (44 min/kg). Fracture properties of the forage used clearly have a strong influence on intake times. The short cut and easily fractured hay used in Study 1 led to very small amounts being extracted per bite leading to very slow intake rates (78 min/kg), and this may also reduce pressure exerted during ingestion. There were also abnormal growing conditions for grass/hay in the summer of 2018 for Study 2, due to lengthy periods of hot, dry weather that impacted growth. It may be that shorter, finer chops encourage more ‘grazing’ behaviour due to the ease of apprehension and limited grams intake per bite, resulting in lower average pull pressures, more bites and longer average intake rates.

The effects of ‘prior’ forage in stable in Study 2 were interesting in relation to palatability and behaviour. Hay was consumed slightly but significantly slower when hay had been available in the stable prior to the test period (as seen in Figure 10). Haylage intake rates were not affected by previous forage in the stable so this points towards lesser palatability or appetite for hay in particular as reported previously [18,19]. The fact that hay was consumed at ‘normal’ rates, even when haylage was in the stable prior to a test, may relate to a ‘change’ in forage effect. Thorne et al. [20], showed an effect of increased intake when providing different forages for selection. LaCasha et al. [21], also showed a decreased preference and net consumption of a forage if previously exposed to it suggesting a positive novelty effect when changing foraging types. So both palatability and novelty—natural multi-species browsing effect may have played a role here. It is important to note that these variables were not a focus of the research and were noted rather than directly manipulated/fully balanced and there was no overall effect (interaction) of prior forage on treatment measures.

When offered hay after previously having haylage in the stable, an increase in mean pull pressures by 37–46% was seen, compared to previously having hay or no forage in the stable. It is possible that this represents learned behaviour, whereby behaviour is remembered from conditions directly previous and applied to the current situation—i.e., higher pressure is required to extract haylage, therefore higher pressure was exerted when haylage was the last forage eaten in the stable, regardless of a change as they are assuming the same requirements will apply. Based on this theory, any acute change in type or manner of forage presented may require an adaptation period, regardless of long term previous experiences within even a short time frame. Findings therefore suggest that previous forage and feeding equipment experience always needs to be controlled through adequate acclimatization periods in further investigations as shown by Ellis et al. [3,6].

## 5. Conclusions

The two studies presented here complement each other but are not identical in methodology. They give a first insight into the pull pressures horses exert when eating from haynets, with one study expanding and building on the previous one. Average pull forces ranged from 74 N (single net, hay) to 156 N (double net, hay) with average maximum forces recorded at 300 N (single, haylage). The majority (80%) of force exerted when eating from haynets was below 70 N (7 kg) for hays used in both studies. For the haylage used in Study 2, 50% of pulls were higher than 50 N with 8% of pulls above 200 N. Considering a horse eating 6 kg of haylage from a net in about 3 h, these pressures may cause some wear on the dental and musculoskeletal system—so for forages highly resistant to apprehension, larger holed haynets may be better. The type and fracture properties of forages used as well as previous forage availability significantly influenced intake behaviour and pull pressures from haynets. Apprehension rate (g/bite) had the greatest impact on feed intake times, with easily fracturing forage leading to more bites and longer intake rates which were further increased by doubling up the haynets. Effects on body posture need to be researched further.

## Figures and Tables

**Figure 1 animals-12-02999-f001:**
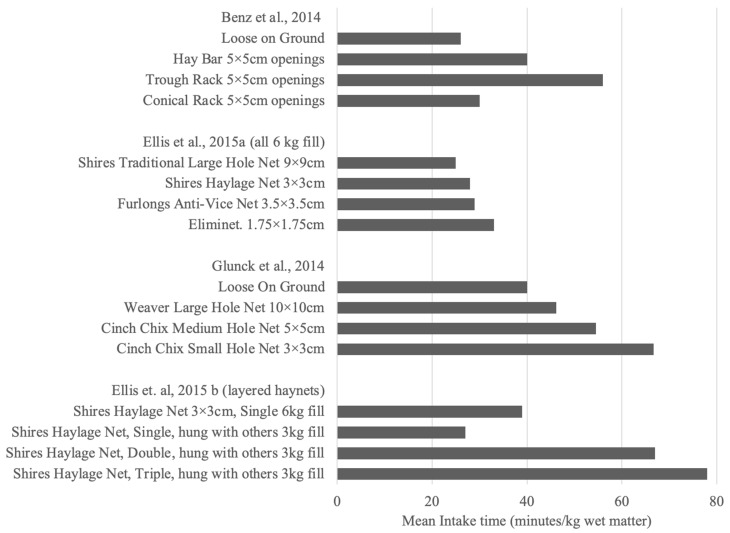
Mean Intake time (minutes/kg wet matter of hay) for horses when eating hay from various different haynet types or similar feeding devices [3,4,6,8].

**Figure 2 animals-12-02999-f002:**
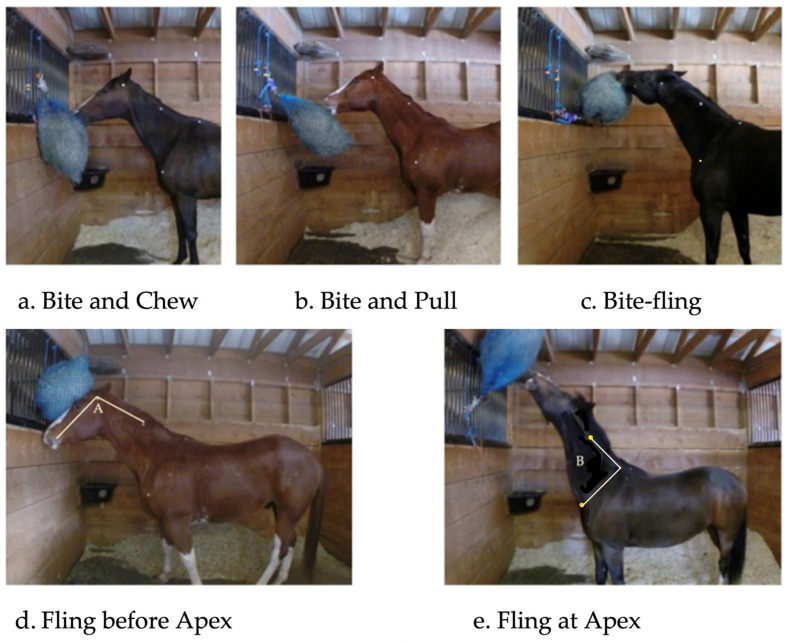
Eating ‘Haynet’ ethogram: (**a**) Eating: bites and chews not easily distinguishable; haynet stays more or less in place; (**b**) distinct bite on haynet and pull away from wall with teeth; (**c**) bite-fling: bite with lifting upwards of haynet; (**d**) fling: push haynet upwards fast with head before taking a bite; (**e**) fling: fling and ‘throw’ of haynet with bite/teeth and head; (Lines: illustration of angles measured A-poll angle, B-wither angle) (Images: Bennett-Skinner).

**Figure 3 animals-12-02999-f003:**
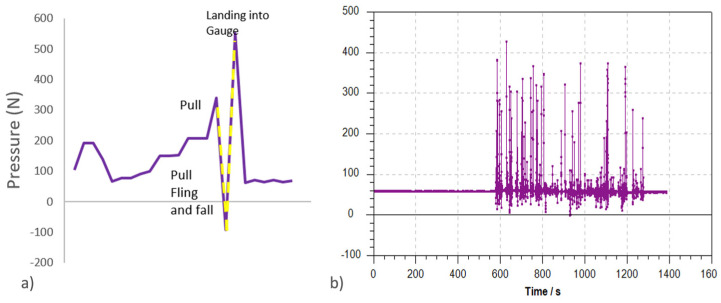
(**a**) Illustration of pressure gauge reading prior to data adjustment showing continued pull slinging up haynet (no pressure to negative push pressure) and sudden drop of haynet as forage is released followed by a sudden high reading as net drops into gauge and; (**b**) Example of corrected pressure readings (once yellow readings from a) have been removed.

**Figure 4 animals-12-02999-f004:**
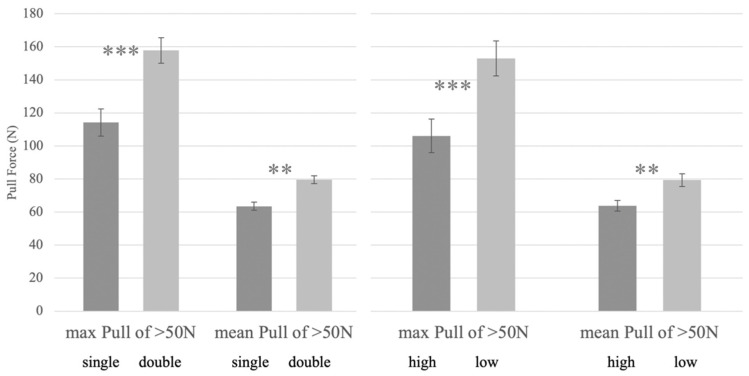
Mean and Mean max pull pressure >50 N recorded for 6 horses eating from either a single or double layered haynet and when hung low or high (s.e. bars shown; *** *p* < 0.001; ** *p* < 0.01; Wilcoxon).

**Figure 5 animals-12-02999-f005:**
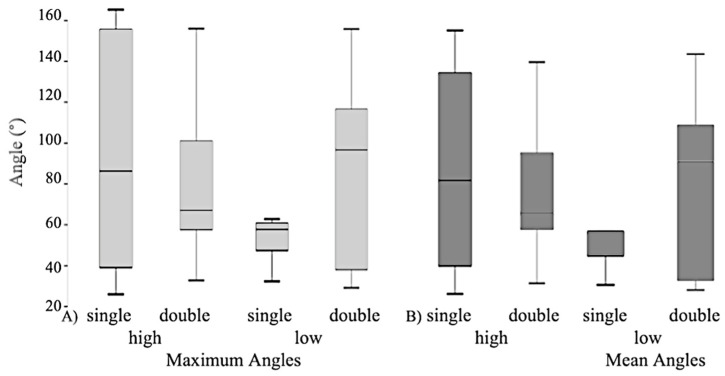
(**A**) Maximum and (**B**) Mean posture angles at the poll (Nose-Poll-Whither) for all non-eating postures (=b—bite and pull; c—bite-fling; d—fling/push).

**Figure 6 animals-12-02999-f006:**
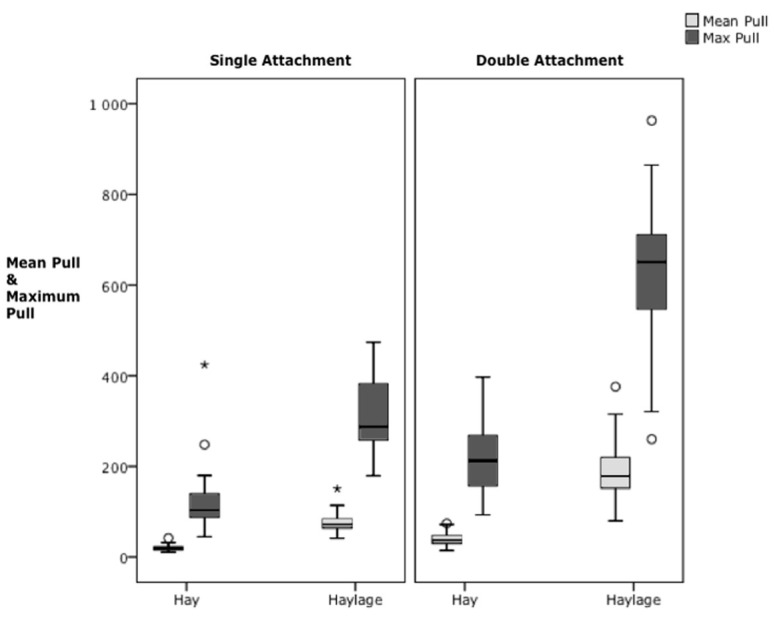
Overall distribution of pull pressure when eating from haynets either from the single attachment or double attachment (mean-grey and maximum-black force) for all horses (n = 10) and all sessions (n = 155), split across the forage treatment type (hay n = 79; haylage n = 76). (Note pressure from double attachment was rigged via a pully system onto the top gauge, making it inclusive of readings from the bottom gauge). All measures within same units differed significantly (Hay single versus Hay double *p* < 0.05; all others *p* < 0.001, Anova, *, ^o^ = outliers).

**Figure 7 animals-12-02999-f007:**
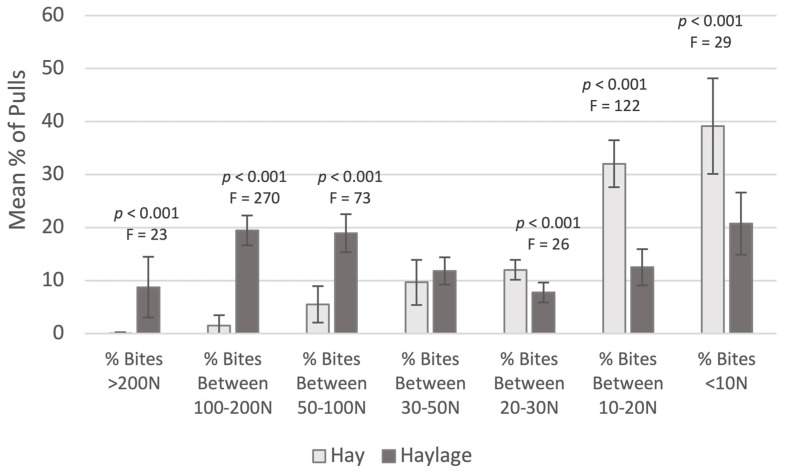
Distribution of bites for hay and haylage for single attachment, according to pressure groups. All segments indicate a significant difference between hay and haylage, apart from bites between 30–50 N (ANOVA; Bonferoni).

**Figure 8 animals-12-02999-f008:**
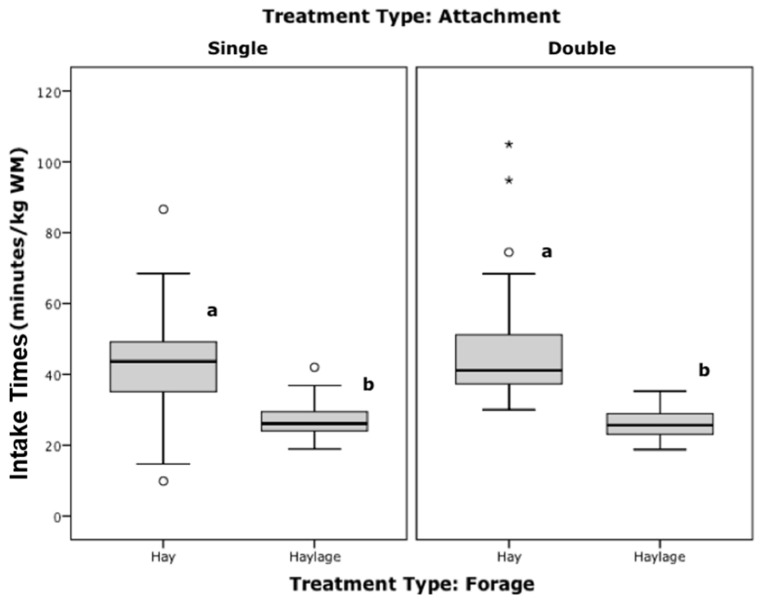
Distribution of intake times, grouped by attachment and forage type. Boxplots that do not share superscripts across all treatments are significantly different (*p* < 0.001, Anova, Bonferroni; WM = wet matter/as fed; *, ^o^ = outliers).

**Figure 9 animals-12-02999-f009:**
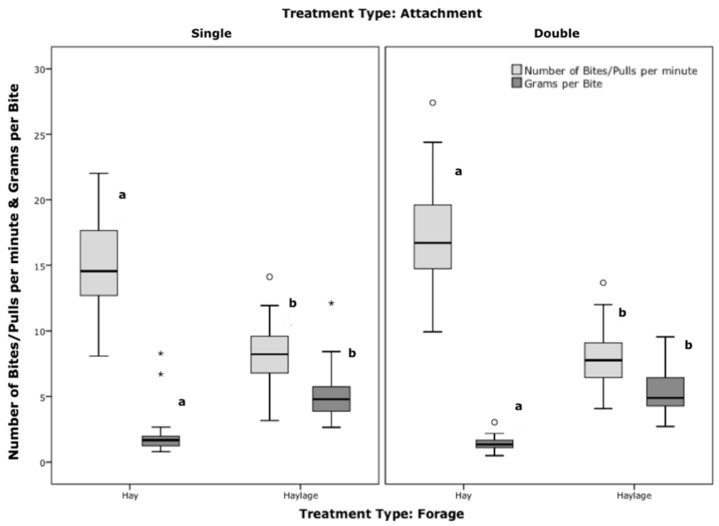
Distributions of number of bites (pulls)/minute (light grey) and grams/ bite (dark grey), grouped by attachment and forage type. Boxplots between forage treatments that do not share lower case superscripts differ significantly (*p* < 0.001, Anova). Boxplots between attachment types but same forage did not show significant differences (*, ^o^ = outliers).

**Figure 10 animals-12-02999-f010:**
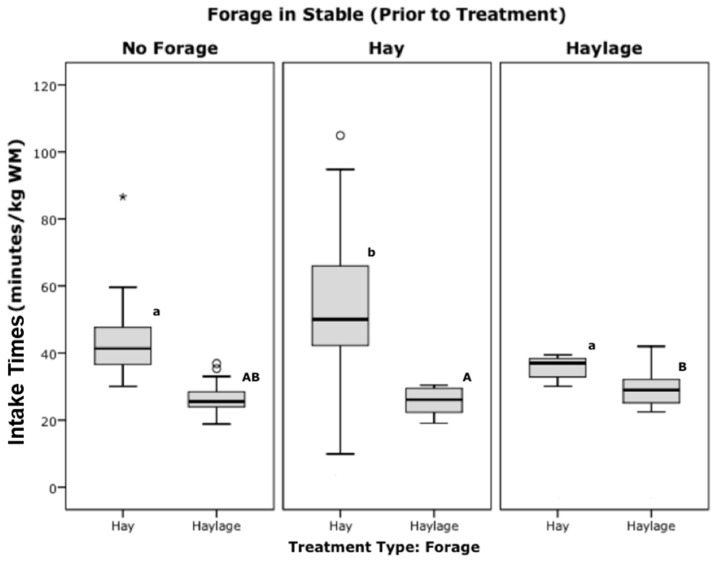
Intake times, grouped by forage in stable prior to treatment and forage treatment type (Significance between hay feeding tests are shown by lower case letters, and between haylage are shown by upper case letters, with those that do not share the same superscripts being significantly different, Anova, Bonferroni, *p* < 0.05; *, ^o^ = outliers).

**Figure 11 animals-12-02999-f011:**
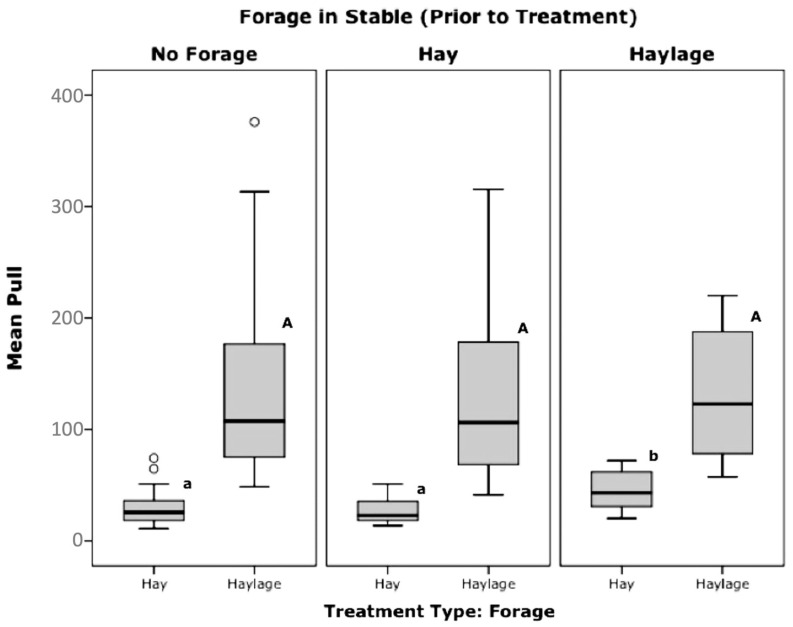
Pull pressures (kPa), grouped by ‘forage in stable prior to treatment’ and forage treatment (Significance between hay feeding tests are shown by lower case letters, and between haylage are shown by upper case letters, with those that do not share the same superscripts being significantly different, Anova, Bonferroni, *p* < 0.05; ^o^ = outliers).

**Figure 12 animals-12-02999-f012:**
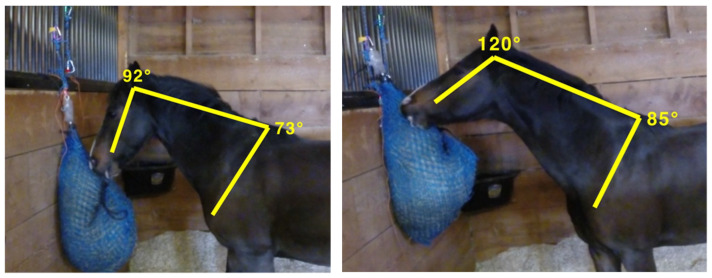
Poll angles recorded when eating from low hung (2.5 cm above withers, **left**) and higher hung haynet (30 cm above withers, **right**) by the same horse. Images illustrate double netted haynets with same fill size.

**Figure 13 animals-12-02999-f013:**
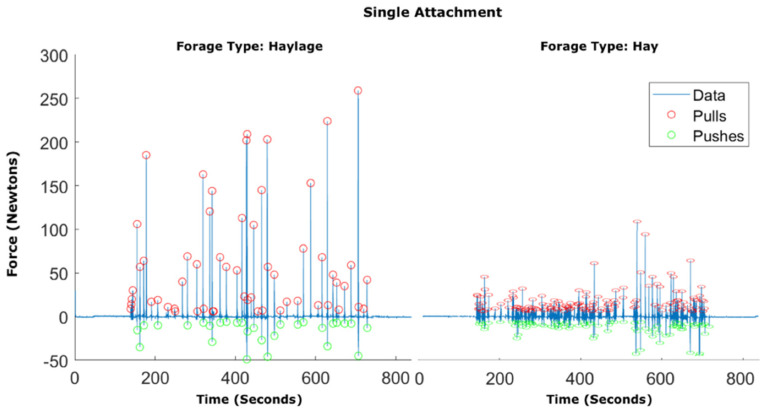
Pressure readings following MATLAB data extraction for one horse eating either haylage (**left**) or hay (**right**) from a single haynet under the single attachment variable.

**Table 1 animals-12-02999-t001:** Organisation of treatment groups by phase and day after randomly assigning horses (n = 6) to either Group A or B (SH = Single layered haynet; DH = Double layered haynet; low & high refers to hanging height).

Test Days	1	2	3	4
	Phase 1			
Group A—SH	SHhigh	SHlow	SHhigh	SHlow
	SHlow	SHhigh	SHlow	SHhigh
Group B—DH	DHhigh	DHlow	DHhigh	DHlow
	DHlow	DHhigh	DHlow	DHhigh
	Phase 2			
Group A—DH	DHhigh	DHlow	DHhigh	DHlow
	DHlow	DHhigh	DHlow	DHhigh
Group B—SH	SHhigh	SHlow	SHhigh	SHlow
	SHlow	SHhigh	SHlow	SHhigh

**Table 2 animals-12-02999-t002:** Organisation of treatment groups by block and day after randomly assigning horses (n = 10) to either Group A or B in a Latin Square Design set up.

Day	1	2	3	4	5	6	7	8
Block 1	HYs	HGs	HYd	HGd	HYs	HYd	HGs	HGd
Group	A	A	B	B	A	A	B	B
Block 2	HYs	HGs	HYd	HGd	HYs	HYd	HGs	HGd
Group	B	B	A	A	B	B	A	A
Block 3	HYd	HGd	HYs	HGs	HGs	HGd	HYs	HYd
Group	A	A	B	B	A	A	B	B
Block 4	HYd	HGd	HYs	HGs	HGs	HGd	HYs	HYd
Group	B	B	A	A	B	B	A	A

HY = hay (92% dry matter), HG = haylage (68% dry matter), s = single attachment, d = double attachment.

**Table 3 animals-12-02999-t003:** Comparison of results Study 1 (Single- SH versus double—DH layered haynets) and Study 2—all single haynets for hay only using (a) Results when removing data <50 N (to align with gauge data from Study 1 for better comparison) and (b) full Results as presented, and one further bite pressure study by Hongo & Akimoto [9] (N = Newtons).

	Mean Pull Pressure (N)	Mean Max Pull Pressure (N)	Intake Time (minutes/kg WM)
Study 1: Hay; 50–5000 N	SH: 74 ± 2.9	SH: 122 ± 6.6	Single: 78 ± 9
	DH: 81 ± 2	DH: 156 ± 9.4	Double: 154 ± 12
Study 2a: 50–2500 N	Hay: 77 ± 2.6	Hay: 123 ± 11.1	Hay: 44 ± 1.8
	Haylage: 135 ± 4	Haylage: 307 ± 18.2	Haylage: 27 ± 1.1
Study 2b; 0–2500 N	Hay: 20 ± 1.8	Hay: 120 ± 13.6	
	Haylage: 74 ± 5.2	Haylage: 307 ± 18.2	
Hongo & Akimoto (2003) Artificial swards		Perennial Rye Grass: 141 ± 11 *	

* Pressure readings only indicative—not directly comparable to Studies 1 & 2 due to different methodology.

## Data Availability

Data will be stored with A.D.E. and The University of Edinburgh.

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
