# Peer review of "Posture and Pull Pressure by Horses When Eating Hay or Haylage from a Hay Net Hung at Various Positions"

_animals, 2022, doi:10.3390/ani12212999_

Round 1

Reviewer 1 Report

Manuscript ID: animals- 1956328

Posture and Pull pressure by horses when eating hay or haylage from a hay net hung at various positions

General comments

The manuscript deals with an interesting topic related to the unnatural feeding positions that horses are forced into when using haynets hung at different heights to provide forage. The current topic is on a topic of relevance and general interest to the readers of the journal. The effect of feeding horses with haynets has been the focus of research in the last years, with some published works. This manuscript presents as a novelty the evaluation of the pressures that horses make when forage is fed into haynets.

The work comprises two studies. The first study assessed head-neck posture and the force exerted on haynets when hung at different heights as either single haynets or doubled-up haynets. The second study assessed pressure when presenting haynets with two different hanging methods (single attachment and double attachment) and with two types of forage (haylage vs hay).

Postural data analysis was only performed in study 1. If we consider the title of the manuscript, there is little information regarding the posture of the horses and the authors report that this data from the 2nd study will be published in another article. Since I was disappointed, the same may be true for other readers.

The manuscript is overall well written, and the trial is well designed. Appropriate statistical analysis is used.

In the results section, some data are not mentioned, which are then presented in the discussion (see specific comments). These results should be shown in the results section. The results are discussed and compared with previous publications.

The paper has several limitations, some identified by the authors, so the conclusions drawn must be interpreted carefully and need to be further validated. Among the limitations are:

- Feed used in the 2 studies was not the same, and the type of forage has an important effect on the intake rate, and in the pressure register to extract forage.

-The amount of food offered in the 2 studies (3kg vs. 6 Kg) was different, which may influence the pressure data obtained

-Pressure readings data were performed by different methods in the 2 studies, and the gauge used in Study 1, did not record low pressure, limiting accurate lower readings and therefore the mean pressures (>50N) in this study may be not the real ones.

- In study 2, the intake behavior data is limited to pressure readings, were not matched to the video recordings, and may not be perfectly representative of real-life bites.

It would be important to point out in the conclusions section, that the conclusions drawn presuppose several limitations stated in the paper.

Please unify the font and size of all figures.

Specific comments

Line 25- Please change to minutes/kg of forage

Lines 77-78- If it is possible, please provide more details concerning the approval of the Ethics Committee where relevant.

Line 87- Please correct: eachp

Line 135- Please check the camera position from the wall

Line 143- Please replace pt., by point

Line 153- Please explain here, the meaning of bite-fling (lifting upwards of haynet) and flinging

Lines 156-160- In the caption of plate 1, put semicolons between the various items.

Line 160- Shoulder angle, is not the correct name for the angle referred in B. In line 143, it was indicated as the wither angle.

Line 207- Please delete “particularly”

Lines 209- 210- Please delete: (most were acclimatized to the forages, others had already 209

received both forages as part of their management). It is already said in the following sentence.

Lines 255-257- Please delete: “The data concerning ‘flinging’ behavior and posture angles as well as matching up pull-bites with video-bites, will be reported in a follow-up paper”.

Line 262- Please do not abbreviate neg.

Line 271- The data is expressed as standard error of the mean, so the abbreviation should be SEM

Line 281- In the Figure 3 chart, statistical differences between groups should be identified. Use asterisks or letters. The P value of the differences should also be written in the figure caption. The same must be done in figures 4 (Line 297) and 5 (Line 311).

Line 293- Please insert (127°±10), after double haynets

Line 296- In figure 4, please use different colors/patterns to distinguish the bars of the A) maximum and B) Mean posture poll angle. Identify statistical differences between groups in the chart and in the figure caption.

Line 298- In the figure 4 caption, please clarify if "a. bite and chew "represents an eating posture. If so, delete it from the legend and leave only the non-eating postures.

Line 300- Insert (figure 4) after “was seen”

Line 304- What’s the meaning of (2)?

Line 311- Please see Figure 5 Y-axis legend. The authors should identify the statistical differences between groups in the graph and in the Figure 5 caption.

Line 338- Please delete figure 6, as Figure 6 refers to the single attached haynets data.

Line 349- Please define the meaning of WM in the figure caption

Lines 351-353- Please rewrite the sentence, since figure 7 corresponds to the intake time data and cannot be referred after figure 8, which corresponds to the bite/minute.

Line 355- Doubt: Were the grams of hay/bite significantly different between the haynets attachment types? In figure 8, for the grams of hay/bite, the capital letters superscripts are different between single haynets (A) and double haynets (B).

Line 363- What does this mean? “it was not possible to influence the type of forage (if present) or the method of presentation in the stable.” Please rewrite.

Line 369- Please write in full what WM means.

Line 378- In the figure 9 caption, indicate how significance is translated. Different letters correspond to significant differences?

Line 388- In the figure 10 caption, indicate how significance is translated.

Line 390- Replace “hay”, with haylage.

Line 429- Please Put a period between “25%” and “this”.

Line 440- Haynet flinging value (2.58 times) does not match the one (and only) indicated in line 291, for the double haynets.

Line 441- Please use uppercase letters for SEM, and delete “here”

Line 456- This plate should be in the results section, and not in the discussion.

Line 457- In plate 2, do both images correspond to the same type of haynets? Please identify if they are with single or double haynets.

lines 484- 499: All the information in lines 484 to 499 (figure 11 and table 2 included) should be in the Results section, as this data was not present in that section

Line 493- 495- The amount of food offered in the 2 studies (3kg vs. 6 Kg) was different, and according to Ellis et al., 2015b, haynet fill seems to influence intake time. In addition, there is a variation in the pressure exerted by the horse, according to the position chosen in the hay net when biting. May this also have affected the pressure data obtained and hence the repeatability of the methodology and the validation of the results between the 2 studies?

Line 498- I don't understand what the repetition of the data from study 2 corresponds to since the table title says they are the results of Study 1 (Single- SH versus double - DH layered haynets) and Study 2 (all single haynets) for hay with a single attachment.

Line 520- Citation was not included and is needed.

Line 560- Please delete (Figure 9).

Line 571- Please delete (Manova).

Author Response

Comments and Suggestions for Authors

General comments

The manuscript deals with an interesting topic related to the unnatural feeding positions that horses are forced into when using haynets hung at different heights to provide forage. The current topic is on a topic of relevance and general interest to the readers of the journal. The effect of feeding horses with haynets has been the focus of research in the last years, with some published works. This manuscript presents as a novelty the evaluation of the pressures that horses make when forage is fed into haynets. 

The work comprises two studies. The first study assessed head-neck posture and the force exerted on haynets when hung at different heights as either single haynets or doubled-up haynets. The second study assessed pressure when presenting haynets with two different hanging methods (single attachment and double attachment) and with two types of forage (haylage vs hay). 

Postural data analysis was only performed in study 1. If we consider the title of the manuscript, there is little information regarding the posture of the horses and the authors report that this data from the 2nd study will be published in another article. Since I was disappointed, the same may be true for other readers.

The manuscript is overall well written, and the trial is well designed. Appropriate statistical analysis is used.

In the results section, some data are not mentioned, which are then presented in the discussion (see specific comments). These results should be shown in the results section. The results are discussed and compared with previous publications. 

The paper has several limitations, some identified by the authors, so the conclusions drawn must be interpreted carefully and need to be further validated. Among the limitations are:

- Feed used in the 2 studies was not the same, and the type of forage has an important effect on the intake rate, and in the pressure register to extract forage.

-The amount of food offered in the 2 studies (3kg vs. 6 Kg) was different, which may influence the pressure data obtained

-Pressure readings data were performed by different methods in the 2 studies, and the gauge used in Study 1, did not record low pressure, limiting accurate lower readings and therefore the mean pressures (>50N) in this study may be not the real ones.

- In study 2, the intake behavior data is limited to pressure readings, were not matched to the video recordings, and may not be perfectly representative of real-life bites.

It would be important to point out in the conclusions section, that the conclusions drawn presuppose several limitations stated in the paper.

Please unify the font and size of all figures.

Specific comments

Line 25- Please change to minutes/kg of forage

Lines 77-78- If it is possible, please provide more details concerning the approval of the Ethics Committee where relevant.

Line 87- Please correct: eachp

Line 135- Please check the camera position from the wall

Line 143- Please replace pt., by point

Line 153- Please explain here, the meaning of bite-fling (lifting upwards of haynet) and flinging

Lines 156-160- In the caption of plate 1, put semicolons between the various items.

Line 160- Shoulder angle, is not the correct name for the angle referred in B. In line 143, it was indicated as the wither angle.

Line 207- Please delete “particularly”

Lines 209- 210- Please delete: (most were acclimatized to the forages, others had already 209

received both forages as part of their management). It is already said in the following sentence.

Lines 255-257- Please delete: “The data concerning ‘flinging’ behavior and posture angles as well as matching up pull-bites with video-bites, will be reported in a follow-up paper”. 

Line 262- Please do not abbreviate neg.

Line 271- The data is expressed as standard error of the mean, so the abbreviation should be SEM

Line 281- In the Figure 3 chart, statistical differences between groups should be identified. Use asterisks or letters. The P value of the differences should also be written in the figure caption. The same must be done in figures 4 (Line 297) and 5 (Line 311).

Line 293- Please insert (127°±10), after double haynets

Line 296- In figure 4, please use different colors/patterns to distinguish the bars of the A) maximum and B) Mean posture poll angle. Identify statistical differences between groups in the chart and in the figure caption.

Line 298- In the figure 4 caption, please clarify if "a. bite and chew "represents an eating posture. If so, delete it from the legend and leave only the non-eating postures.

Line 300- Insert (figure 4) after “was seen”

Line 304- What’s the meaning of (2)?

Line 311- Please see Figure 5 Y-axis legend. The authors should identify the statistical differences between groups in the graph and in the Figure 5 caption.

Line 338- Please delete figure 6, as Figure 6 refers to the single attached haynets data.

Line 349- Please define the meaning of WM in the figure caption

Lines 351-353- Please rewrite the sentence, since figure 7 corresponds to the intake time data and cannot be referred after figure 8, which corresponds to the bite/minute.

Line 355- Doubt: Were the grams of hay/bite significantly different between the haynets attachment types? In figure 8, for the grams of hay/bite, the capital letters superscripts are different between single haynets (A) and double haynets (B).

Line 363- What does this mean? “it was not possible to influence the type of forage (if present) or the method of presentation in the stable.” Please rewrite.

Line 369- Please write in full what WM means.

Line 378- In the figure 9 caption, indicate how significance is translated. Different letters correspond to significant differences?

Line 388- In the figure 10 caption, indicate how significance is translated. 

Line 390- Replace “hay”, with haylage.

Line 429- Please Put a period between “25%” and “this”.

Line 440- Haynet flinging value (2.58 times) does not match the one (and only) indicated in line 291, for the double haynets.

Line 441- Please use uppercase letters for SEM, and delete “here” 

Line 456- This plate should be in the results section, and not in the discussion.

Line 457- In plate 2, do both images correspond to the same type of haynets? Please identify if they are with single or double haynets.

lines 484- 499: All the information in lines 484 to 499 (figure 11 and table 2 included) should be in the Results section, as this data was not present in that section

Line 493- 495- The amount of food offered in the 2 studies (3kg vs. 6 Kg) was different, and according to Ellis et al., 2015b, haynet fill seems to influence intake time. In addition, there is a variation in the pressure exerted by the horse, according to the position chosen in the hay net when biting. May this also have affected the pressure data obtained and hence the repeatability of the methodology and the validation of the results between the 2 studies?

Line 498- I don't understand what the repetition of the data from study 2 corresponds to since the table title says they are the results of Study 1 (Single- SH versus double - DH layered haynets) and Study 2 (all single haynets) for hay with a single attachment.

Line 520- Citation was not included and is needed.

Line 560- Please delete (Figure 9).

Line 571- Please delete (Manova).

Thank you very much for your comments and in depth suggestions. We have taken these carefully into consideration. all the replies match up to your comments but there is no way for us to upload an additional word document so we had no choice but to copy and paste. Apologies if this means you need to re-match comments. 

Yes, it is a shame we have only been able to assess posture angles for study 1 so far but nevertheless results show a good collection of postural behaviours at various treatments and we would therefore retain that aspect in the title.  The second study built upon the results of the first study which could be seen as a preparatory study, however the depth of data collection does make this valid even with only 6 horses. We have amended the aim slightly in line 65 -67 to ‘downgrade’ the importance of posture to these studies.

This has been addressed further below, although we do not feel that there is a point in completely replicating results in the result and discussion section when discussing them – therefore results which arose from critically evaluating the basic results through comparison and amalgamation had been put in the discussion for ease of reading/understanding.

Indeed, thank you

We have addressed that in the discussion – that was partially the point of the follow up study and has been discussed in depth – results show some consistency and we using various feedstuffs actually adds to our understanding

Again – as above – we have highlighted that strongly in the discussion – one study builds upon the other… and this again could be seen as a strength in terms of adding to our understanding on the influence of forage properties.

We have addressed that clearly in the discussion by leaving lower pressure readings out for direct comparison between the studies allowed for direct comparison in the discussion – this is partially why we wanted the studies to be presented together in one paper.

The first study focused on maximum pressure whereas  the results from the second provide accurate mean readings that are transferable to real life impact, and by adjusting the data to directly compare with study 1, we illustrated replicability of the methodology.

Yes, we agree that ideally we would have liked to have more time to do this. We have addressed this by developing software to remove non-bite readings – which was aligned with video recordings in a small limited comparison – we aim to analyse all video footage in depth further in the future. 

We have added this, thank you and in terms of all comments above we added a further short introduction to the discussion section as well.

Checked – 1 x sizing amended

We are not sure where this change should be implemented – we added hay to the figure 1 title

We added a note

Done, thank you

indeed that was confusing, thanks - amended

done

Amended

Amended

Amended

Deleted

Deleted

Deleted

Corrected

Corrected

This is not compulsory if the data are given in the text but we have added these to Fig 3

For the boxplots in Fig 4 and 5 the text clearly explains the relevant significance and the figures give additional break up of data  adding all combinations of significance levels here would look confusing and the boxplots speak for themselves. We added explanation to Fig 5

thank you we moved this and – we added difference in angle to further underline this

Colouration amended – the rest explained above

Removed

Moved

removed

Clarified differences in title

Ok – clarified – added Figures 5-6 earlier and changed end to Figure 5 

Clarified

Thank you – indeed figure numbers got confused here – corrected and amended

Thank you, indeed – we corrected this and amended text

Changed to control…and expanded

this abbreviation has been explained several times previously so it is not nec. To write out in full every time – it is similar to kg a common abbreviation  – we added additional line to methods

Amended

Clarified

Clarified

Corrected

Amended to match

Amended

We feel this is for illustrative purposes and has relevance in this section as it is used to discuss how findings from study 1 influenced the design of study 2 – we would not normally include single horse results in the result section

Clarified

This section does not include new results and is based on data shown above, it rather discusses and compares results that have been stated in the results section, and once again Figure 11 from a single horses is used to underpin the discussion rather than present results.

Table 2 represents a ‘coming together’ of both studies and one other study which attempted to measure pressures exerted. We agree with your comments and have added the following sentence in the main text: As with any study comparison, we are not comparing an exact replication of methods but d-spite a difference in forages used and haynet fill between the two studies ( which may well have influenced results somewhat as highlighted by Ellis et al., 2015b), the pressures exerted when pulling hay or haylages from the haynet show some overall consistency.

A number of discussed factors mean that results from both studies have been, and should be, treated as independent results, but we feel the similarities in findings and the attempt to standardise methodology is still supportive of repeatability and validity.

This is explained in the sentence above but indeed needs to be added to the title – we amended that, thanks.

There was a mistake in numbers there as well which may add to confusion that has been amended

Apologies – slipped through from an earlier version Added citation, reference and to discussion point

 this shows the data we are discussing? So why not refer back to it?

Deleted

Reviewer 2 Report

A well-written study and the first of its kind evaluating the posture and pull pressure by horses when eating hay or haylage from a hay net hung at various positions. The study design is appropriate, the methods are very well described. The results are clearly presented and support the conclusion of the study. The study is very timely and relevant for equine practitioners, horse owners, students and researchers.

Author Response

 Thank you for the novel concept and a nice article. Please consider the following edits.

Line 29-32: Formatting issue. Italics should not be used for this text.

Line 71-72: How did you arrive at the sample size of six horses? Can you generalize your findings? Because the sample size is insufficient. And what kind of horses were they? Their breed?

Line 71: What kind of work did these horses do? Race? Polo? Dressage? What was the size of these horses? Please consider these changes to improve the presentation of your study material.

Line 71: What was the health status of these horses, as this could have an impact on pulling ability, particularly dental health? Were the eating habits and appetites of the horses in both studies similar?

Line 120: Please provide information about your equipment (camera specifications) and the frames per second of the videos you used for analysis.

 Line 125-132: Interesting research design. I was wondering how the friction between the haynets and the wall affects the force meter readings.

142-145: Because you used images to analyze angles. How did you learn about a specific frame of video (image) in which the horse was pulling or exerting maximum force?

484-485: “Mean pull pressure was 2.6 times higher and maximum pull pressure was 1.6 times higher when eating haylage than when eating hay, but hay led to a 83% increase in bite frequency.” What could be the cause of this? And if your horse has a natural preference for one type of food over another. What effect will this have on his pull?

Many thanks!

Amended

Variation of pressures between horses has never been measured before so a calculation on sample grouping could not be used. This was seen as a good intial size to assess initial pressure ranges and variabilities and influences of these on a fairly homogenous group of horses in size and experience. It formed the basis for further research in the form of Study 2 whith a greater sample population, which would then provide a better basis for generalisability and future research. Specific breeds were not recorded, but information felt to be impactful to findings such as age, health, and weight were collected and in both studies some individuals excluded.

In Study 1 horses were used for general leisure riding at light worklevel. Study 2, it is discussed that horses partook of low level and low frequency schooling exercise as part of students lessons at the college. We added some detail to methods.

To be included in both studies, horses were required to be healthy and have no underlying clinical management that would be considered impactful to study results. All horses had recent dental checks and none had any adverse findings or conditions that would have impacted results. Horses were accustomed to forage types and kept within similar/same routines, though monitoring of prior forage conditions did take place in Study 2

All this is presented within methods.

added – frames per second unknown – for study 1 a still picture of each bit at max angle was taken as well as still pictures for every 10 sec as described in methods.  Study 2 not relevant as this data is not presented

This would certainly be a good basis for future research in this area, and build upon current findings

explained above and in methods

Fracture properties are discussed and considered to be highly influential to these findings, as the hay results in small gram extractions that likely necessitated and increase in bite frequency in order to increase the gram amount received. Whereas the haylage being longer cut and potentially because of higher moisture content, large gram amounts were extracted but higher force required to achieve the extraction. In previous forage experience there was an attempt to monitor this as an effect and it is suggestive that horses did interact slightly  differently with forages based on other factors unrelated to fracture quality, as discussed. 

Reviewer 3 Report

Thank you for the novel concept and a nice article. Please consider the following edits.

Line 29-32: Formatting issue. Italics should not be used for this text.

Line 71-72: How did you arrive at the sample size of six horses? Can you generalize your findings? Because the sample size is insufficient. And what kind of horses were they? Their breed?

Line 71: What kind of work did these horses do? Race? Polo? Dressage? What was the size of these horses? Please consider these changes to improve the presentation of your study material.

Line 71: What was the health status of these horses, as this could have an impact on pulling ability, particularly dental health? Were the eating habits and appetites of the horses in both studies similar?

Line 120: Please provide information about your equipment (camera specifications) and the frames per second of the videos you used for analysis.

Line 125-132: Interesting research design. I was wondering how the friction between the haynets and the wall affects the force meter readings.

142-145: Because you used images to analyze angles. How did you learn about a specific frame of video (image) in which the horse was pulling or exerting maximum force?

484-485: “Mean pull pressure was 2.6 times higher and maximum pull pressure was 1.6 times 484 higher when eating haylage than when eating hay, but hay led to a 83% increase in bite 485 frequency.” What could be the cause of this? And if your horse has a natural preference for one type of food over another. What effect will this have on his pull?

Author Response

(The authors gave the same response as above.)
